# Normal β-Cell *Glut2* Expression Is not Required for Regulating Glucose-Stimulated Insulin Secretion and Systemic Glucose Homeostasis in Mice

**DOI:** 10.3390/biom13030540

**Published:** 2023-03-16

**Authors:** Siresha Bathina, Tumininu S. Faniyan, Lauren Bainbridge, Autumn Davis, Kavaljit H. Chhabra

**Affiliations:** 1Department of Medicine, Division of Endocrinology, Diabetes and Metabolism, School of Medicine and Dentistry, University of Rochester, Rochester, NY 14642, USA; 2Department of Medicine, Division of Endocrinology, Diabetes and Metabolism, Baylor College of Medicine, Houston, TX 77030, USA; 3Department of Pharmacology and Physiology, University of Rochester Medical Center, Rochester, NY 14642, USA

**Keywords:** β-cell GLUT2, mouse model, glucose, insulin secretion

## Abstract

Objective: Glucose transporter 2 (GLUT2) is expressed in the pancreatic β-cell, intestine, liver, and kidney in mice. Although GLUT2 is considered as a major regulator of insulin secretion, in vivo contribution of β-cell *Glut2* to glucose-stimulated insulin secretion and systemic glucose homeostasis is undefined. Therefore, the main objective of this study is to determine the role of β-cell *Glut2* in regulating insulin secretion and blood glucose levels in mice. Methods: We produced mice in which we can knock down *Glut2* at a desired time specifically in β-cells (β-*Glut2* KD) by crossing *Glut2*^LoxP/LoxP^ mice with *Ins1^CreERT2^* mouse strain and using the Cre-Lox recombination technique. We measured fasting blood glucose levels, glucose tolerance, and glucose-stimulated insulin secretion in the β-*Glut2* KD mice. We used qRT-PCR and immunofluorescence to validate the deficiency of β-cell *Glut2* in β-*Glut2* KD mice. Results: We report that both male and female β-*Glut2* KD mice have normal glucose-stimulated insulin secretion. Moreover, the β-*Glut2* KD mice exhibit normal fasting blood glucose levels and glucose tolerance. The β-*Glut2* KD mice have upregulated GLUT1 in islets. Conclusions: Our findings demonstrate that normal β-cell *Glut2* expression is not essential for regulating glucose-stimulated insulin secretion and systemic glucose homeostasis in mice. Therefore, the currently assumed role of β-cell GLUT2 in regulating insulin secretion and blood glucose levels needs to be recalibrated. This will allow an opportunity to determine the contribution of other β-cell glucose transporters or factors whose normal expression may be necessary for mediating glucose stimulated insulin secretion.

## 1. Introduction

Glucose transporter 2 (GLUT2) is a major transporter of glucose in the β-cells, liver, intestine, and kidney in mice. GLUT2 deficiency in humans causes Fanconi–Bickel syndrome [1], which is characterized by glycogen accumulation, glycosuria, impaired glucose and galactose tolerance, and nephropathy. Loss of *Glut2* in the intestine in mice improves glucose tolerance [2]. Deficiency of hepatic GLUT2 contributes to progressive impairment of glucose regulation in mice [3]. Moreover, we recently reported the contribution of renal GLUT2 to systemic glucose homeostasis [4,5,6]. Loss of renal *Glut2* protects mice from streptozocin induced hyperglycemia and diet induced obesity [4].

Global *Glut2* knockout mice [7] exhibit hypoinsulinemia, hyperglycemia, and die within three weeks after birth. In contrast, humans with GLUT2 deficiency show good prognosis [8], although they may experience transient neonatal diabetes [9,10]. Humans and rodents have differential expression of β-cell GLUT2 [11]. GLUT2 is considered as a major glucose transporter in rodent β-cells [12], but this is not true for human β-cells [11]. Whether normal β-cell *Glut2* expression is required for regulating glucose-stimulated insulin secretion and systemic glucose regulation in adult mice is unknown.

In this study, we determined the contribution of β-cell *Glut2* to glucose stimulated insulin secretion and systemic glucose homeostasis in mice. We used an inducible Cre-Lox system to produce β-cell GLUT2 deficiency at a desired time in adult mice. This strategy minimized potential developmental defects that would otherwise affect the phenotype in mice because of absence of *Glut2* during prenatal and perinatal periods.

## 2. Materials and Methods

### 2.1. Animal Care and Generation of Mice

All mouse procedures were approved by the Institutional Animal Care and Use Committee at the University of Rochester and were performed according to the US Public Health Service guidelines for the humane care and use of experimental animals. The reporting in the manuscript follows the recommendations in the ARRIVE guidelines. The mice were housed in ventilated cages under controlled temperature (~23 °C) and photoperiod (12 h light/dark cycle, lights on from 06:00 h to 18:00 h) conditions with free access to Hydropac water (Lab Products, Seaford, DE, USA) and regular laboratory chow (5010, LabDiet, Arden Hills, MN, USA). The mice were randomly assigned to different experimental groups. We used two mouse lines with their littermate controls in this study: *Glut2*^LoxP/LoxP^ and *Ins1^CreERT2^* mouse strains [4,13].

We generated *Ins1^CreERT2^*;*Glut2*^loxP/loxP^ mice and their littermate controls by first crossing the *Glut2*^loxP/loxP^ mice [4] with *Ins1^CreERT2^* mouse line [13] (026802, The Jackson Laboratory, Bar Harbor, ME, USA) followed by a second generation of intercrossing between *Ins1^CreERT2^*;*Glut2*^loxP/+^ and *Glut2*^loxP/loxP^ mice. To knockdown β-cell *Glut2* in *Ins1^CreERT2^*;*Glut2*^loxP/loxP^ mice, we injected them with tamoxifen (100 mg/kg dissolved first in 1 part 100% *v/v* ethanol by incubating the mixture at 60 °C for 20 min followed by addition of 9 parts of sesame oil, T5648, and S3547, Sigma, St. Louis, MO, USA) once daily for five consecutive days. The control mice were also injected with tamoxifen using the same protocol.

### 2.2. Islet Isolation Procedure

We isolated islets for measuring mRNA levels of *Glut2* to confirm its deficiency in β-cell *Glut2* knockdown mice. We inflated the mouse pancreas by injecting 2 mL of Liberase TL (Roche 05401020001, 1.3 Wünsch units/mL in RPMI 1640 medium) using the bile duct, as described previously [14,15]. We collected the inflated pancreata and incubated them at 37 °C in a water bath for 14 min to digest the tissue. We then followed published protocols [14,15] to separate and purify the islets. The isolated islets were immediately frozen in liquid nitrogen and stored at −80 °C in a freezer until the mRNA assay.

### 2.3. Oral Glucose Tolerance Test

After fasting mice for 6 h (08:00–14:00 h), their tails were slightly nicked laterally using a razor blade, and a drop of blood was used to measure baseline (0 min) blood glucose with AlphaTRAK 2 glucometer. Then, a fixed dose of glucose (100 mg/mouse in 250–300 µL water, G8270; Sigma) was administered into each mouse by oral gavage (18 g needle, FNS-18-2, Kent Scientific Corporation, Torrington, CT, USA) and blood glucose levels were determined at 15, 30, 60, and 120 min after the oral glucose administration.

### 2.4. Glucose Stimulated Insulin Secretion

To determine glucose-stimulated insulin secretion, we fasted mice for 6 h (08:00–14:00 h) and collected tail blood using heparinized capillary tubes (22-362-566, Fisherbrand) at baseline (0 min) and 20 min after glucose administration (100 mg glucose dissolved in 250–300 µL water, oral gavage). The blood was centrifuged for 20 min at 2000× *g* speed and 4 °C to separate plasma. Plasma insulin levels were measured using ELISA (90080; Crystal Chem, Elk Grove Village, IL, USA).

### 2.5. RT-qPCR

We used Aurum Total RNA Fatty and Fibrous Tissue Kit (7326830, Bio-rad, Hercules, CA, USA) to extract total RNA. Five hundred ng total RNA and random hexamer primers (1708891, iScript cDNA synthesis kit, Bio-Rad) were used to generate cDNA. RT-qPCR was performed using a StepOne Real Time PCR System (Applied Biosystems) and SYBR green master mix (1725124, Bio-Rad). We used the following primers, *Glut2* (*Slc2a2*): 5′-GAA GGA ACT CAG TAC AGC AGT G-3′ and 5′-TCA TCC ACA TTC AGT ACA GGA C-3′; *Glut1* (*Slc2a1*): 5′-GTG GTG AGT GTG GAT G-3′ and 5′-AGT TCG GCT ATA ACA CTG GTG-3′; *Hprt*: 5′-AAC AAA GTC TGG CCT GTA TCC-3′ and 5′-CCC CAA AAT GGT TAA GGT TGC-3′. All primers were used at a final concentration of 500 nM. The relative quantity of each mRNA was calculated from standard curves and normalized to the internal control *Hprt*, and then normalized to the mean of corresponding controls.

### 2.6. Immunohistochemistry

After euthanizing the mice, their pancreata were collected and fixed in 10% formalin before embedding them in paraffin. The samples were cut into 5µm sections using a microtome and placed onto slides. Before staining the sections, they were deparaffinized and rehydrated using xylene (3 min incubation, twice), ethanol (3 min incubation each with 1:1 with xylene followed by 100%, 70%, and 50% ethanol in water), and water (5 min under running tap water to rinse off ethanol). Antigen retrieval was carried out by incubating the sections in sodium citrate buffer (pH 6.0) in water bath at 95 °C for 25 min. Nonspecific staining was blocked by subsequent incubation with 10% goat serum in tris-buffered saline with 0.1% Tween 20 detergent (TBST) for 60 min at room temperature. The tissue sections were then processed for GLUT1, GLUT2, or insulin staining using previously validated antibodies [4,5]: rabbit anti-GLUT1 (1:250 dilution in TBST, ab652 or ab115730, Abcam, Waltham, MA, USA), rabbit anti-GLUT2 (1:250 dilution in TBST, 600-401-GN3, Rockland Immunochemicals, Pottstown, PA, USA), or mouse anti-insulin (1:100 dilution in TBST, sc-8033, Santa Cruz Biotechnology, Inc., Dallas, TX, USA) incubated at 4 °C overnight. The next day, the sections were washed three times (10 min/wash) in TBST and then incubated with secondary goat anti-rabbit or anti-mouse antibody conjugated to Alexa Fluor 488 or 647 as appropriate (ab150077, ab150115; Abcam; 1:1000 dilution in TBST) for 2 h at room temperature. After this incubation, the sections were washed three times (10 min/wash) in TBST and counterstained with DAPI (D9542; Sigma; 1 μg/mL) for 1 min. This was followed by a quick wash with TBST again before the sections were air-dried and coverslipped using ProLong Antifade mounting medium (P36930; Molecular Probes, Waltham, MA, USA). Images were captured using Keyence fluorescence microscope BZ-X800. We quantified the images using the NIH ImageJ software, version 1.53n (https://imagej.nih.gov/ij/download.html, accessed on 19 April 2022 through 9 June 2022).

### 2.7. Statistical Analyses

We included 6 to 12 mice per group in this study based on our previously published reports [4,5]. All data are presented as mean ± SEM and were analyzed by Two-tailed Student’s paired or unpaired *t* test, one-way ANOVA or two-way ANOVA followed by a Bonferroni post hoc multiple comparison test when appropriate. All analyses were performed using Prism 8.0 (GraphPad, San Diego, CA, USA) and differences were considered statistically significant at *p* value < 0.05.

## 3. Results

### 3.1. Generation and Validation of β-cell Glut2 Knockdown Mice

We generated *Ins1^CreERT2^*;*Glut2*^LoxP/LoxP^ mice and their littermate controls by crossing our *Glut2*^LoxP/LoxP^ mouse line [4] with *Ins1^CreERT2^* mouse strain [13] (Figure 1A). This *Ins1^CreERT2^* mouse line is validated for tamoxifen inducible Cre-Lox recombination selectively in β-cells [13]. We injected the *Ins1^CreERT2^*;*Glut2*^LoxP/LoxP^ mice with tamoxifen (100 mg/kg, once daily for five days) to produce β-cell *Glut2* knockdown (β-*Glut2* KD) mice. We measured mRNA and protein levels of islet or β-cell GLUT2 using qRT-PCR and immunofluorescence, respectively, one week after the last injection of tamoxifen or after completion of the study. We observed that β-*Glut2* KD mice have about 15% GLUT2 in islets or β-cells compared to their littermate controls (Figure 1B,C), demonstrating the deficiency of *Glut2* and therefore, validating the mouse model. As we did not observe a complete knockout of β-cell *Glut2* in our mouse model describe here, we used the term knockdown for the level of *Glut2* deficiency achieved in the model.

### 3.2. β-Cell Glut2 Knockdown Mice Have Normal Glucose-Stimulated Insulin Secretion

Although GLUT2 is considered as a major regulator of glucose-stimulated insulin secretion, it is unknown whether normal expression of β-cell GLUT2 is necessary for regulating this phenomenon. To address this question, we measured glucose-stimulated insulin secretion four and eight weeks after inducing β-cell GLUT2 deficiency in 8-week old male and female β-*Glut2* KD mice. We observed that fasting plasma insulin levels and insulin secretion in response to glucose administration in the *Glut2* knockdown mice were similar to that observed in their littermate control mice (Figure 2). The variability in the extent of β-cell *Glut2* knockdown achieved in β-*Glut2* KD mice could have contributed to the observed wide range of glucose-stimulated insulin secretion. Nevertheless, these results demonstrate that normal expression of β-cell *Glut2* is not essential for regulating glucose-stimulated insulin secretion in mice.

### 3.3. β-Cell Glut2 Knockdown Mice Have Normal Glucose Tolerance

As global *Glut2* knockout mice display impaired glucose homeostasis [7], we determined whether deficiency of β-cell *Glut2* in adult mice recapitulates this defect. We performed oral glucose tolerance tests four and eight weeks after inducing β-cell GLUT2 deficiency in 8-week old male and female β-*Glut2* KD mice. The mice have normal glucose tolerance (Figure 3A–D) despite the deficiency of β-cell *Glut2*. Moreover, β-*Glut2* KD mice have 6 h fasting (Female, 169.8 ± 7 vs. 166.3 ± 8.6; Male, 189.5 ± 9 vs. 190 ± 4 mg/dL, Control vs. β-*Glut2* KD mice, n = 6) and random non-fasting (Female, 206 ± 14 vs. 213 ± 11; Male, 217 ± 9 vs. 223 ± 16 mg/dL, Control vs. β-*Glut2* KD mice, n = 6) blood glucose levels similar to that observed in their littermate control mice. As some reports have suggested the contribution of other glucose transporters such as GLUT1 to mediating glucose transport and metabolism in absence of GLUT2 [16,17], we measured protein levels and gene expression of islet *Glut1* using immunofluorescence and qRT-PCR in female mice. We found that islet *Glut1* was increased in female β-*Glut2* KD mice compared to their littermate controls (Figure 3E,F). This increase was predominantly in insulin-positive cells (β-cells) as quantified using ImageJ software (Figure 3E). Overall, these findings indicate that normal level of β-cell GLUT2 is not necessary for regulating systemic glucose homeostasis.

## 4. Discussion

In this study, we determined the contribution of β-cell *Glut2* to glucose stimulated insulin secretion and systemic glucose homeostasis in adult mice. We observed that β-cell *Glut2* knockdown mice have similar insulin secretion as their littermate controls in response to glucose administration. Moreover, the knockdown mice show normal fasting blood glucose levels and glucose tolerance compared to their littermate control mice. These findings demonstrate that normal expression of β-cell GLUT2 is not necessary for regulating systemic glucose homeostasis in adult mice.

GLUT2 is regarded as a major regulator of insulin secretion. For example, almost all textbooks [18] and reviews [12] on this topic depict β-cell GLUT2 as a carrier through which glucose is transported into β-cells to trigger subsequent events and accomplish glucose stimulated insulin secretion. This knowledge is based on results obtained from either global *Glut2* knockout mice [7] or in vitro studies using isolated islets [19]. As *Glut2* is absent during prenatal and perinatal periods in global *Glut2* knockout mice, some of their phenotype could be attributed to developmental defects arising from the *Glut2* deficiency. For example, during the pancreas development, β-cells are produced from *Glut2*-expressing cells [20], which may influence insulin secretion in the global *Glut2* knockout mice. To determine the contribution of β-cell *Glut2* to glucose-stimulated insulin secretion and glucose homeostasis independently of potential developmental defects, we induced knockdown of β-cell *Glut2* in adult mice using the Cre-lox system.

Our findings here demonstrate that deficiency of β-cell GLUT2 does not affect glucose-stimulated insulin secretion and blood glucose levels. In contrast, loss of hepatic *Glut2* progressively impairs glucose homeostasis in mice [3]. Moreover, deficiency of *Glut2* in the intestine [2] or kidney [4] improves glucose tolerance in mice by decreasing absorption or increasing excretion of glucose, respectively. These findings indicate that glycosuria observed in global *Glut2* knockout mice is attributed to loss of renal *Glut2*. Hypoinsulinemia and short life span observed in global *Glut2* knockout mice could be due to developmental defects or secondary to lack of glucose absorption and profound loss of glucose in urine. It is important to note that transgenic re-expression of β-cell *Glut2* (at embryonic stage) is sufficient to normalize fed blood glucose levels and glucose stimulated insulin secretion in otherwise global *Glut2* knockout mice [16]. In humans, genetic GLUT2 deficiency causes Fanconi–Bickel syndrome, occurrence of which is rare. Unlike the *Glut2* knockout mice, overall prognosis of this syndrome in humans is good [8] with normal fertility [21,22]. The syndrome may manifest temporary neonatal diabetes [9,10,23]. Altogether, normal level of β-cell GLUT2 is not essential for regulating glucose-stimulated insulin secretion and systemic glucose homeostasis in mice.

Some studies have reported that β-cell GLUT1 can maintain glucose transport and metabolism in absence of β-cell GLUT2. For example, β-cell GLUT1 is sufficient to reverse defects observed in global *Glut2* knockout mice or in vitro β-cells [16,17]. Our present study supports these findings as we observed an increase in islet GLUT1 in the β-*Glut2* KD mice, which may be adequate for maintaining normal glucose transport and metabolism including glucose-stimulated insulin secretion despite the deficiency of β-cell *Glut2*. This speculation is corroborated by a previous report [24] suggesting that β-cell GLUT2 is not necessary for regulating glucose homeostasis in mice. These findings indicate that other glucose transporters compensate for the deficiency of β-cell GLUT2. The cross-talk between the transporters may be explained by transcription factors such as HNF1α, which are involved in regulating the gene expression of such transporters. For example, we recently reported that renal GLUT2 deficiency downregulates renal SGLT2 via HNF1α [4].

This study has some limitations. Given that GLUT2 is a high capacity low affinity glucose transporter, it is possible that remnant β-cell *Glut2* in the β-cell *Glut2* knockdown mice may be sufficient to maintain normal systemic glucose homeostasis and glucose stimulated insulin secretion. We used the term knockdown instead of knockout because about 15% β-cell *Glut2* is still present after the Cre-Lox recombination in our mouse model described here. We did not determine the consequence of β-cell GLUT2 deficiency on glucose metabolism or glycolysis in β-cells. Although we observed that islet GLUT1 is upregulated in β-*Glut2* KD mice, it is unclear whether this increase is responsible for maintaining normal glucose stimulated insulin secretion and glucose homeostasis in these mice. Moreover, other glucose transporters such as SGLT1 or SGLT2 are not expressed in β-cells per previous reports [25,26], therefore we did not determine their contribution to glucose homeostasis in this present study. We measured the effects of β-cell *Glut2* deficiency on glucose homeostasis until eight weeks after inducing the *Glut2* deficiency. Therefore, secondary effects on glucose homeostasis arising beyond eight weeks after the knockdown of β-cell *Glut2* cannot be excluded.

In summary, β-cell GLUT2 deficiency in mice does not impair their glucose stimulated insulin secretion and blood glucose levels. Therefore, the theory of β-cell GLUT2-mediated insulin secretion needs to be revisited to further appreciate the necessity of normal expression of other factors or transporters that are required for glucose-stimulated insulin secretion.

## Figures and Tables

**Figure 1 biomolecules-13-00540-f001:**
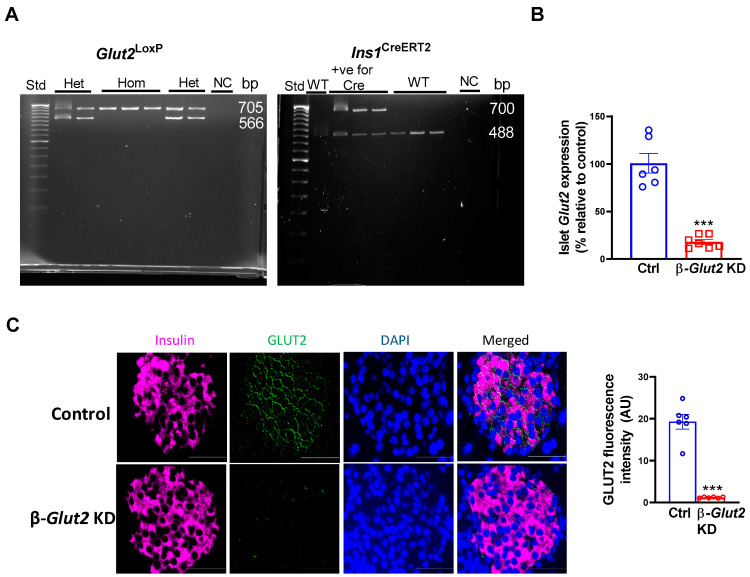
Generation and validation of β-cell *Glut2* knockdown (β-*Glut2* KD) mice. Representative genotypes of mice used in this study (**A**). Het, heterozygous; Hom, homozygous for *Glut2*^loxP^. Std, standard DNA ladder; NC, negative control; bp, base pair. Results from qRT-PCR showing knockdown of islet *Glut2* in 8 weeks old male β-*Glut2* KD mice one week after inducing β-cell *Glut2* deficiency, n = 6 or 7 (**B**). Immunofluorescence demonstrating deficiency of β-cell GLUT2 in male β-*Glut2* KD mice, n = 6 or 7 (**C**). Three islets per section and 3 sections per mouse were quantified (**C**). Scale bar is 50 µm. Ctrl, control *Glut2*^loxP/loxP^ mice; AU, arbitrary unit. Student’s unpaired *t*-test was used for comparisons, *** *p* < 0.001. Error bars are mean ± SEM.

**Figure 2 biomolecules-13-00540-f002:**
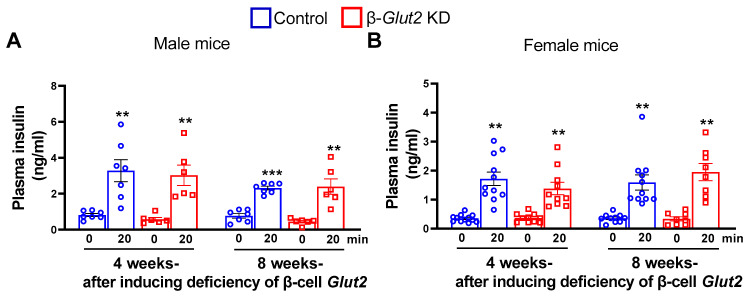
β-cell *Glut2* knockdown (β-*Glut2* KD) mice have normal glucose-stimulated insulin secretion. Fasting (6 h) plasma insulin levels (0 min) and glucose-stimulated insulin secretion (20 min) in male, n = 6 or 7 (**A**) and female, n = 8, 10, or 11 (**B**) mice 4 and 8 weeks after inducing β-cell *Glut2* deficiency. The mice were 8–10 weeks old when β-cell *Glut2* deficiency was induced. Repeated measures two-way ANOVA followed by Bonferroni’s multiple comparison test were used for comparisons. ** *p* < 0.01, *** *p* < 0.001 vs. their corresponding baseline (0 min) groups. Error bars are mean ± SEM.

**Figure 3 biomolecules-13-00540-f003:**
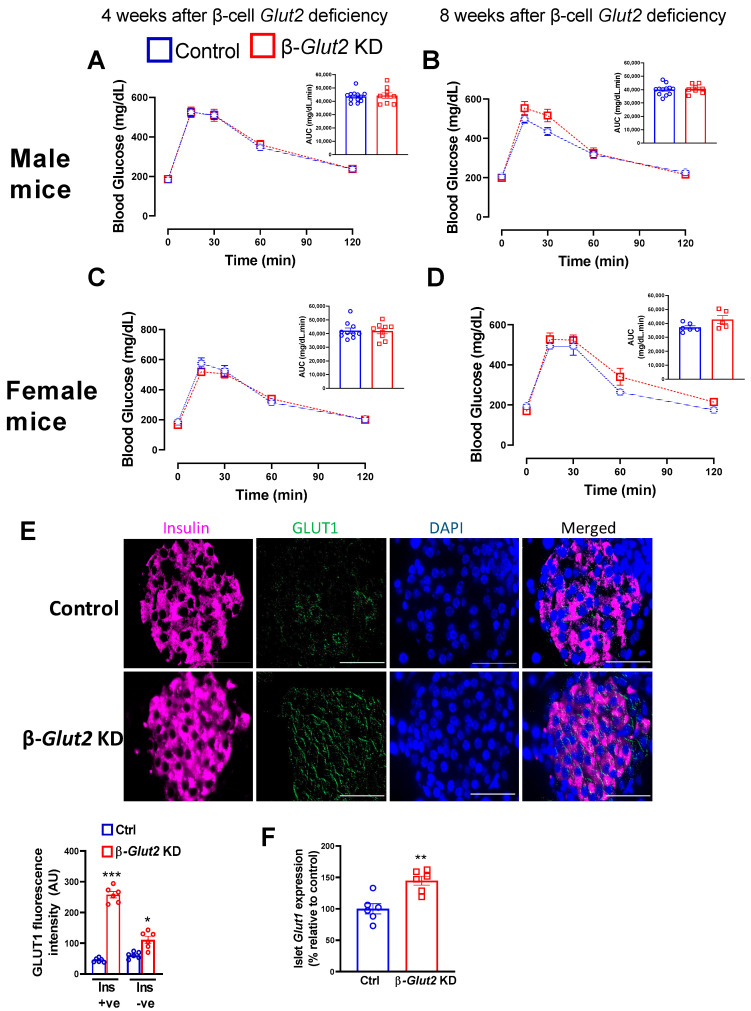
β-cell *Glut2* knockdown (β-*Glut2* KD) mice exhibit normal glucose tolerance. Results obtained from oral glucose tolerance tests in male, n = 9 or 12 (**A**,**B**) and female, n = 6–10 (**C**,**D**) mice 4 (**A**,**C**) and 8 (**B**,**D**) weeks after inducing β-cell *Glut2* deficiency. Area under the curve (AUC) is shown in inset bar graphs. The mice were 8–10 weeks old when β-cell *Glut2* deficiency was induced. Immunofluorescence, n = 6 (**E**) and qRT-PCR, n = 6 (**F**) demonstrating upregulation of islet Glut1 in female β-*Glut2* KD mice after completion of the study. 3 islets per section and 3 sections per mouse were quantified (**E**). Ins +ve and −ve, quantification of GLUT1 fluorescence in insulin positive and negative staining. Scale bar is 50 µm. Control, *Glut2*^loxP/loxP^ mice; AU, arbitrary unit. Student’s unpaired *t*-test was used for comparisons, * *p* < 0.05, ** *p* < 0.01, *** *p* < 0.001. Error bars are mean ± SEM.

## Data Availability

All data are available in the main text. The reagents and mouse model used in this study are available via material transfer agreement addressed to the corresponding author.

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
