# Peer review of "Normal β-Cell Glut2 Expression Is not Required for Regulating Glucose-Stimulated Insulin Secretion and Systemic Glucose Homeostasis in Mice"

_biomolecules, 2023, doi:10.3390/biom13030540_

Round 1

Reviewer 1 Report

The authors have submitted a well structured manuscript, exploring the role of the GLUT2 transporter in beta cells in mice.   The limitations of the study are well articulated, although the translatable impact of the study is clearly debatable.  The following queries/comments pertain:

1. The authors need to be careful in specifying that the study relates specifically to mice (e.g. concluding paragraph, opening sentence, of the introduction).

2. Sp. Line 67; where % is employed, w/v or v/v should be specified

3. It is not clear how many mice were in each group, or how that decision was arrived at: did the authors perform power calculations?

4. With regard to Q-PCR: were levels of the housekeeper Hprt invariant during the study?

5. Line 145: a comparator is needed

6. Figure 1 legend: the number of mice, versus samples analysed, needs to made explicit; it is not always possible to assess this from the symbols shown.

7. Line 159: Period not required after beta

8. Figure 2: the numbers of mice analysed seem to differ from Figure 1: what is the rationale for this?  And the authors should comment on the degree of variance on the data - what is the reason for this?  Can it be related to the extent of knockdown of Glut2?

9. The induction of Glut1 reported in the text: the data supporting this contention derive from female mice only, but this is not what is implied in the text in the Results or Discussion.   The mechanism by which Glut2 kd may induce Glut1 expression: the authors may wish to suggest some possibilities here.

Reviewer 2 Report

This is timely and important study that examines and clarifies the importance of normal GLUT2 expression for glucose homeostasis and insulin secretion in vivo. Using the Cre/Lox inducible model, and beta-cell specific deletion approach, the authors directly tested and revisited the commonly accepted obligatory role of GLUT-2 transporter in regulating insulin and glucose serum levels. The manuscript provides concise but important new information about the role of this transporter in the global glucose homeostasis. I have only a few, minor comments and suggestions to the authors in effort to improve the clarity and impact of this interesting manuscript.

Ln. 165: Authors disclosed normal fasting plasma insulin in their KO mice without providing any quantitative data for comparison.

Fig. 3: It will be informative if the authors also provide steady state fasting and non-fasting glucose serum levels. More importantly, results of the GTT test performed under non-fasting conditions, if available, should be added as they would reinforce main conclusions of this study.

Fig. 1C and Fig. 3E: immunofluorescence data should be analyzed and reported as % change in GLUT2 and GLUT 1 expression in beta- (insulin positive) and other (insulin negative) islet cells. This would provide information on how selective and how efficient knockdown of Glut2 gene in transgenic islet cells has been achieved (Fig. 1). It may provide explanation for residual GLUT2 immunoreactivity in Cre/Lox-edited islets. It may also help with discerning the cellular origin of the proposed GLUT 1 compensatory effect on GSIS described in Fig. 3E ( i.e. if the proposed compensatory effect via GLUT1 overexpression originates predominantly within beta-cells or other islet cells may be involved).

Round 2

Reviewer 1 Report

The authors have addressed the concerns raised, and I would recommend publication of this paper